# Effect of the Microsand Fraction on the Ballistic Resistance of UHP-SFRC

**DOI:** 10.3390/ma15175916

**Published:** 2022-08-26

**Authors:** Přemysl Kheml, Kristýna Carrera, Pavel Horák

**Affiliations:** Faculty of Civil Engineering, Czech Technical University in Prague, Thákurova 7, 166 29 Prague, Czech Republic

**Keywords:** penetration resistance, projectile impact, high-speed loading, ballistic resistance, UHP-SFRC, fibres

## Abstract

This work investigates the effect of various sand fractions on the ballistic resistance of ultra-high-performance steel-fibre-reinforced concrete (UHP-SFRC) samples. We specifically investigated replacing expensive and generally inaccessible microsands with commonly available sands. The tests and the measured values show that replacing part of the microsand with the more commonly and economically acceptable 0/2 mm aggregate fraction minimises the resulting mechanical properties and ballistic resistance. The most common type of ammunition was used to test all sample bodies, which is a 7.62 × 39 mm calibre with an all-metal jacket and a mild steel core. The damage’s extent and mode were determined using a 3D scanner operating on photogrammetry.

## 1. Introduction

This work evaluates the effect of fine sands on the ballistic resistance of the resulting mixture. Specifically, this is a comparison of commonly used sands that are used for normal strength concrete (NSC) and microsands are used for ultra-high-performance concrete (UHPC). Microsands are a non-negligible part of the final price of UHPC, and their industrial availability is very limited. It is therefore important to find a way to appropriately replace microsands with commonly used sands. This work asks the question of to what extent conventional sands can replace microsands and how much this replacement affects the ballistic resistance of the resulting material.

Proper optimisation of the composite mixture, considering its financial efficiency and, in this case, ballistic resistance is an essential condition for practical application [1,2]. Good optimization of the raw material of this type of silicate composite thus enables its greater applicability in industry. To simplify this subtask, emphasis was placed on finding the most suitable grain size curve. An elemental blend composition was chosen, where in addition to microsand, other components, such as admixtures and additives, were left untreated in terms of their proportional representation in the mixture. One of the essential prerequisites for high-performing concretes’ correspondingly high compressive strength is their compactness and integrity. One of the main ways to achieve this attribute is the use of aggregates or fillers with grain sizes of up to 2 mm [3]. In this case, the grain sizes and commonly used sand fractions are around 4 to 8 mm, and some manufacturers are able to offer a fraction of 0/2 mm. For this reason, they are used more frequently for high-strength and ultra-high-performance concretes, where the grain size of technical and glass sands ranges from 0.1 to 1.6 mm. A significant disadvantage of these fine sands is their high purchase price.

The factor that influences the depth of penetration (DOP) is the size and type of aggregate used. From a global perspective, concrete is considered a homogeneous material, with its heterogeneous internal structure and composition determining the macroscopic properties and parameters that are further used in the design and calculations. In terms of the internal structure, the load-bearing element with the highest density is the aggregate, which, compared to the other components, also has a significantly higher strength. An optimised concrete mix and the correct use of aggregate strength can significantly reduce the penetration depth of a projectile. As a projectile passes through the specimen, it propagates along the straightest possible trajectory, even at the cost of a more significant loss of kinetic energy, as is the case for crack formation and propagation under impact loading [4]. The projectile does not have enough time to find the most energetically favourable path. Hence, it has to traverse even aggregate grains with significantly higher strength and density than the surrounding matrix and pores. However, this failure mode consumes significantly more kinetic energy and results in deformation and an increase in the contact area of the projectile [5].

The projectile’s kinetic energy is consumed at the front of the specimen to compress the material and cause subsequent damage, which is characterised by the fragmentation of the material and its subsequent release in the opposite direction to that of the projectile. The resulting crater has the approximate shape of a coma cone. The projectile continues to penetrate deeper into the sample and forms a cylindrical tunnel with a diameter equal to the size of the projectile. The second stage is named the ‘tunnelling section’ after the characteristic tunnel-like shape of the breach. When the critical penetration value is exceeded, the tensile or shear strength is exceeded, and initial cracks and fragments form in the rear of the sample. This is also significantly influenced by the pressure wave that is already generated at the initial contact between the projectile and the specimen, which is reflected at the rear and contributes considerably to the tensile stress. This third phase, in which a cone-shaped crater is formed on the rear side of the specimen, again directly adjacent to the tunnel-shaped part, is referred to as “rear cratering” [6,7,8,9].

## 2. Materials and Methods

### 2.1. Experimental Campaign

The following test procedure was proposed to compare the effect of replacing microsands with commonly used sands. A mixture of UHP-SFRC containing only microsands (Section 2.1.1) was selected. These microsands were gradually replaced by commonly used sand (Section 2.1.2). The production of samples from the proposed mixtures is described in Section 2.1.3. Ballistic tests were carried out using a 7.62 × 39 mm bullet, described in detail in Section 2.2. The samples were evaluated using photogrammetry and a 3D scanner (Section 2.3).

#### 2.1.1. Reference Samples of UHP-SFRC

All mixtures were based on the same aggregate or sand reference mix and weight ratio by changing only the types of individual fractions and the types of materials. The reference mixture always contained a quick-setting cement of class 52.5 R, the same amount of silica fume flour, superplasticizer, and dispersed fibre reinforcement in the form of short steel fibres. The reference mixture was developed by the Experimental Center of the Czech Technical University in Prague in previous research. The water/cement ratio was set at 0.24. In this low ratio, part of the binder does not hydrate, and the unhydrated binder then acts as a filler. The number of steel fibres was also determined based on previous research. The volume of content was derived from previous studies [10,11,12] as an optimal quantity concerning the price, workability, and resulting performance. The composition of the reference mix is given in Table 1.

#### 2.1.2. Samples with Altered Sand Contents

The essential prerequisites for high-performance concrete’s corresponding high compressive strength are its compactness and integrity. One of the main ways to achieve this attribute is by using aggregates or fillers with grain sizes of up to 2 mm [3]. In this case, the grain sizes and commonly used sand fractions are around 4 to 8 mm, and some manufacturers are able to offer a fraction of 0/2 mm. For this reason, technical and glass sands are more commonly used for high-strength and high-performance concretes, where grain sizes range from 0.1 to 1.6 mm as a standard. A significant disadvantage of these fine sands is their high purchase price. In this work, six composite mixes with different gradation curves, fractions, types of aggregates, and sands were prepared (Table 2).

The detailed composition of the mixture is the same as that presented in Section 2.1.1, with the only variation being the choice of aggregate or the composition and granulometry of the sands and microsands. Table 3 shows each mix’s proposed weight ratios per kg of aggregate and sands. Reference mixture A contained only microsands with 0.1 to 1.2 mm fractions. Mixtures B–F also contained aggregates in the 0–2 mm fraction as a cheaper substitute for microsands. The type and amount of aggregates and microsands for mixing B–F were chosen to keep the granulometry similar to that of mixture A. (Figure 1). For mixtures B and C, only one fraction of microsand and an aggregate fraction of 0/2 mm were used. Mix D and Mixes E–F contained two and three fractions of microsand and aggregate fraction 0–2, respectively.

The graph in Figure 1 shows the distribution of the granulometry content of the individual mixtures as a function of grain size. The subpoints relative to each other are interconnected, thus showing a simplified envelope of the fractions contained per 1 kg of aggregate. The graph shows some disproportion in the grain content distribution in mixture A, which consists only of microsand and contains the highest amount of grains in the interval of 0.3 mm to 0.75 mm. The lowest average value of the other curves in this interval was 73.3% for mixture A, while mixture B differed by only 11.1% in the 0.7 mm fraction. In the other cases where the aggregate fraction of 0/2 mm was also used, the curves have a rounder shape, i.e., they have higher grain contents in the extreme intervals from 0.15 mm to 0.25 mm and from 0.75 mm to 3.0 mm. The highest difference was found for the 0.9 mm and 1.12 mm fractions, which were 57.6% and 31.6% of the average for the mixture A value of the other distributions.

#### 2.1.3. Sample Production

The same production procedure was followed for all mixes. The fine ingredients, i.e., microsilica, cement, silica fume, and different types of sands and aggregates, were first placed in the mixer. All ingredients were homogenized together for 5 min. Subsequently, water and superplasticizers were added, and the mix was agitated for 5 min. The water dosage was always regulated, considering the rheological properties and boundary conditions such as the moisture of the mixer wall, aggregate moisture, etc. In the third step, steel fibres were added and homogenized for a further 5 min. Steel fibres were dosed into the mix at a rate of 1.5% of the mix volume. Due to its good workability, the mix was designed to be self-compacting, so it could be poured into the moulds without the need to further vibrate the mix. The test specimens for the ballistic tests were 300 mm × 400 mm and 50 mm thick. The samples were left in the formwork for 24 h after pouring and then, after depotting, stored in containers of water at 20 °C for 27 days. Beams measuring 40 mm × 40 mm × 160 mm were used to determine the mechanical parameters. These beams were deliberately used to correlate as closely as possible with the results of the ballistic tests, where the test specimens had a thickness of 5 cm. The use of standard beams with dimensions 100 mm × 100 mm × 400 mm would not consider the so-called “wall effect” or the so-called “size effect”, which can affect the measured values by more than 15% [13]. The specific procedure for testing these solids can be found in the cement testing standard CSN EN 196-1 [14].

### 2.2. Ballistic Resistance Testing

All tested samples were first anchored to a unique steel frame (Figure 2), ensuring the sample’s stability during impact and the correct directional orientation of the sample to the shooter. All samples were attached to the frame using rectification screws that were placed 50 mm apart in the corners of the frame. This mounting ensured an even distribution of forces on the specimens. A schematic of the ballistic test is shown in Figure 3. The shooter and the specimen fixed in the frame had 20 m between them. Approximately 1 m from the shooter, optical gates were placed to measure the velocities of the projectiles [11,12,15,16]. The average measured velocities ranged between 680 and 720 m/s. According to Kneubuehl [17], the impact velocity of the projectile was 22 m/s lower than its muzzle velocity. Ammunition with an all-metal jacket, a steel core with a yield strength of 550 MPa, and a lead penetrator was used for all samples. The projectile weight of the ogival shape with its core and jacket was 8.04 g. The projectile length was 26.7 mm, and the projectile diameter was 7.62 mm (Figure 4). The test was chosen to test the semi-automatic rifle type, the civilian equivalent of the military SA-58 rifle. The 7.62 × 39 calibre ammunition used was also a civilian type. 

### 2.3. Sample Analysis

After the projectile hit the sample, an approximately conical crater was formed in the material. The craters were scanned using a 3D scanner that works on the principle of multiple photogrammetry. Figure 5 shows the actual damage to the sample, and Figure 6 is a 3D scan. The principle of multiple photogrammetry is based on taking multiple images, with every two images overlapping or at least part of them. Due to this overlap, it is possible to determine and calculate the spatial coordinates and subsequently create a 3D model [18]. 

## 3. Results

### 3.1. UHP-SFRC Samples

#### 3.1.1. Mechanical Parameters of Samples

The average values of the mechanical properties of the UHPSFRC samples are given in Table 4, Table 5, Table 6, Table 7, Table 8 and Table 9, according to the individual substrates. The tensile flexural strength (f_cf_) and compressive strength (f_c_) were measured 1, 7, and 28 days after placing the mixture in the moulds. The density of the tested samples was also tabulated. The mechanical parameters were measured on beams with dimensions of 40 mm × 40 mm × 160 mm. The spacing of the supports in the three-point tensile test of bending the beams at the tension in bending was 100 mm. The samples were loaded with a DSM2500 hydraulic press-100 and a ZUZ-200 with controlled deformation capabilities and maximum forces of 2500 kN and 200 kN, respectively, for compression and tension.

The tables and figures show that the mixture with the highest compressive strength was mixture A, with an average compressive strength of 146.9 MPa. Mixture A contained only microsands with fractions from 0.1 to 1.2 mm. The mixture with the lowest average compressive strength was mixture B, with a strength of 104.7 MPa. Together with mixture C, these were the mixtures with the highest proportions of aggregate fraction of 0/2 mm. Replacing the microsand reduces the resulting compressive strength of the mixture.

#### 3.1.2. Ballistic Parameters of Samples

All craters created by the projectile impacts were scanned using the SLS-2 David 3D scanner, which works on the principle of multiframe photogrammetry. The resulting data were then evaluated using the DAVID Laserscanner. DAVID directly processed the resulting data used in this work, so there was no need to process the data in another program.

## 4. Discussion

The observed values of the mechanical parameters are given in Section 3.1.1 (Table 4, Table 5, Table 6, Table 7, Table 8 and Table 9). A comparison of these values can be seen in the following graphs (Figure 7 and Figure 8). It can be seen in the graph that the highest compressive strengths (f_c_) after 28 days were reached by mixture A, which contained only microparticles, with a value of 146.9 MPa. On the other hand, the lowest value of 104.7 MPa was achieved by mixture B, which contained a substitute in the form of an aggregate fraction of 0/2 mm and microsand with a fraction of 0.3 to 0.8 mm. A similar compressive strength was achieved by substituting C, 109.8 MPa, which contained, in addition to aggregates of the 0/2 fraction, 0.1 to 0.6 mm. In contrast, mixes D, E, and F, which contained more types of fractions, recorded compressive strengths of 121.9 MPa, 136.3 MPa, and 129.8 MPa, respectively. The results show the dependence of compressive strength on the grain size curve, with a more optimal and uniform distribution of fractions, i.e., limiting the use of one type of microsand or one fraction results in an increase in the compressive strength by up to 23%. The lowest flexural tensile strength values were achieved by mixes B and C, at 22.7 MPa and 23.5 MPa, respectively. However, the highest values were 33.9 MPa for substitute F and 29.1 MPa for reference substitute A, which can be attributed to the high degree of homogeneity and cohesion of the matrix with fibre reinforcement.

The damage sizes of the samples according to the individual dimensions are given in Table 10. Considering the standard deviations, the observed depth of penetration (DOP) did not show a significant increasing or decreasing trend, with all measured values ranging from 22.6 to 25.5 mm (Figure 9). The difference in these values of 2.9 mm corresponded to less than 6% of the total thickness of the tested samples. On the contrary, an increasing linear trend can be seen in the case of the dependence of the entrance crater diameter on the compressive strength (Figure 10). With increasing strength, the diameter of the entrance crater increased from 52.6 mm at 104.7 MPa to 63.0 mm at 146.9 MPa. The difference between these two extreme values was 10.4 mm at 42.2 MPa, which means a difference of 16.5% in the diameter of the crater and 28.7% in compressive strength. In this case, it is necessary to consider the relatively small standard deviations, which, although they did not exceed more than 5% of the overall average, amounted to almost 27% in terms of the difference between the highest and lowest values. A similar increasing trend is evident in the case of the size of the area of the entrance crater (Figure 11). Here, the linear increase in the crater surface area with increasing compressive strength is clearly seen. The size of the entrance crater areas ranged from 4011 mm^2^ at a compressive strength of 104.7 MPa to a value of 5072 mm^2^ at 146.9 MPa. In terms of growth, the difference between the two extreme intrusions was 1061 mm^2^ at 42.2 MPa or an increase in the area of 21% with an increase in strength of less than 29%. This trend only confirms the effect of the compressive strength on the magnitude of the damage and, in general, the resistance of the composite to projectile impact [5,19,20]. The increased size of the crater with increasing compressive strength can be attributed to the composite’s higher mechanical energy absorption capacity. For similar material compositions, a composite with higher compressive strength can absorb a greater amount of projectile energy, resulting in more significant damage in the form of a larger entry crater [21]. A relatively similar increasing trend is evident in the exit crater area and flexural tensile strength (Figure 12 and Figure 13). In the interval of up to 25 MPa, the increase in area size was significantly higher, from 6712 mm^2^ at 22.7 MPa to 7828 mm^2^ at 24.0 MPa. Therefore, this was an increase of 1116 mm^2^ with an increase in flexural strength of 1.3 MPa. The growth in the other three mixes was only 900 mm^2^ with an increase of 9.9 MPa. This phenomenon can again be attributed to the higher energy absorption in the rear of the specimen, resulting in a larger exit crater after the projectile passes through. The behaviour described above on the entrance and exit sides of the crater corresponds to deformation models of thin-walled structures [6,7,8,9].

## 5. Conclusions

The tests and the measured values show that replacing part of the microsand with the more commonly and economically acceptable 0/2 mm aggregate fraction minimises the resulting mechanical properties and ballistic resistance. In some cases, higher values were achieved in tensile flexural strength and lower penetration depth values (Table 10). In general, the mechanical parameters depend on the correct distribution of aggregate and microsand fractions or the most optimised curve of higher-order polynomial grain size (Figure 1). The most suitable parameters were achieved using three types of microsand (microsand 01/06, microsand 03/08, and microsand 06/12) and an aggregate fraction of 0/2 mm. On the contrary, the lowest mechanical values were found for the mixture using a type of microsand (microsand 03/08). The composition of the reference mix (mix A) also shows that the reduction in the volume of the sand fractions with grain sizes between 0.9 and 6.0 mm does not have a significant negative effect on the resulting mechanical properties of the composite. It has been shown that, for the UHP-SFRC, microsand can be replaced by commonly used sand without significantly affecting the ballistic properties. For a statistically accurate expression of the dependence of ballistic resistance on the sand fraction, the number of samples needs to be increased. Therefore, the authors have not reported this dependence in this paper.

## Figures and Tables

**Figure 1 materials-15-05916-f001:**
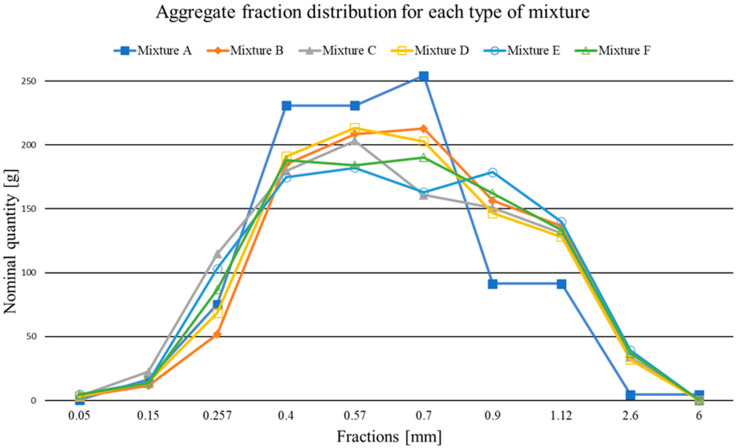
Aggregate fraction distribution for individual mixtures. The x-axis shows the mean values of the fraction intervals reported by the manufacturer.

**Figure 2 materials-15-05916-f002:**
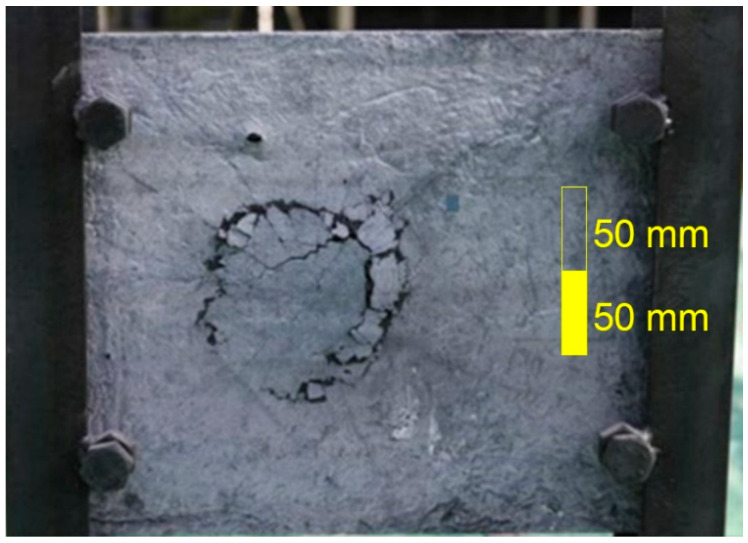
A unique steel frame with the sample.

**Figure 3 materials-15-05916-f003:**
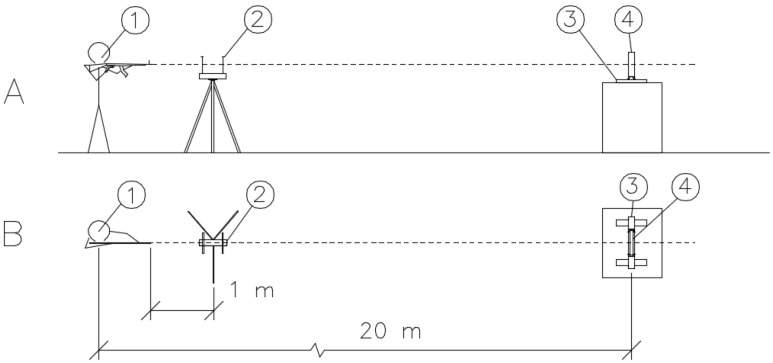
The layout of the experiment: (**A**) side view and (**B**) plan view. 1. Shooter, 2. Velocity gates, 3. Sample mounting frame, 4. Sample.

**Figure 4 materials-15-05916-f004:**
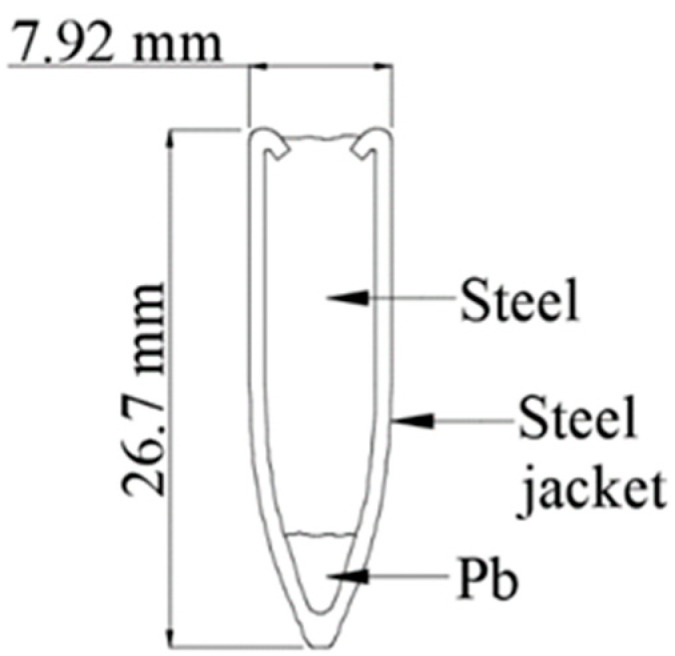
Projectile cut.

**Figure 5 materials-15-05916-f005:**
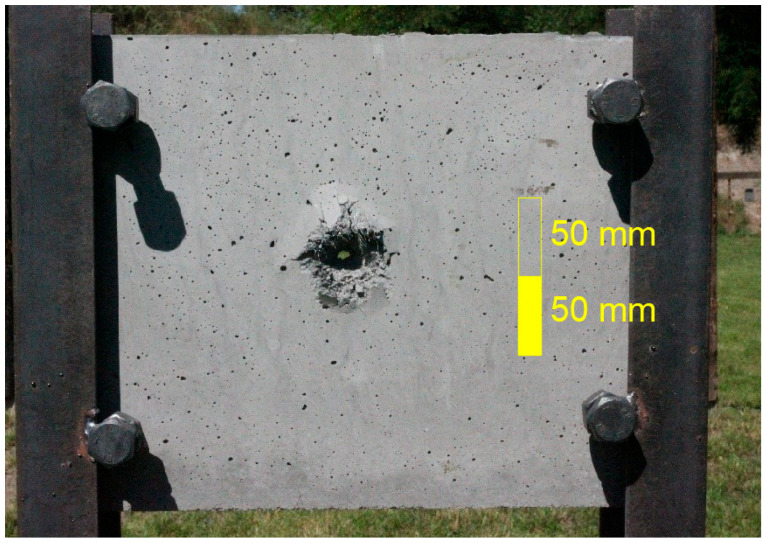
Sample after ballistic test.

**Figure 6 materials-15-05916-f006:**
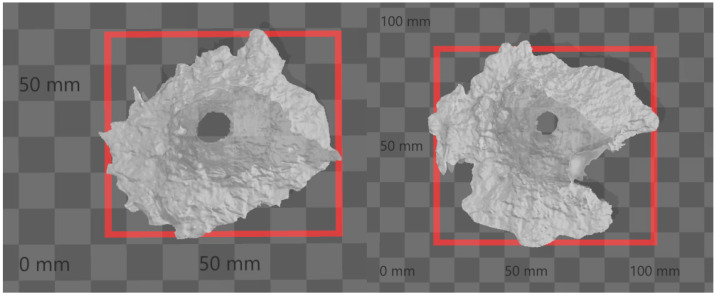
Three-dimensional scan of the crater. **Left**—entrance side, **right**—exit side.

**Figure 7 materials-15-05916-f007:**
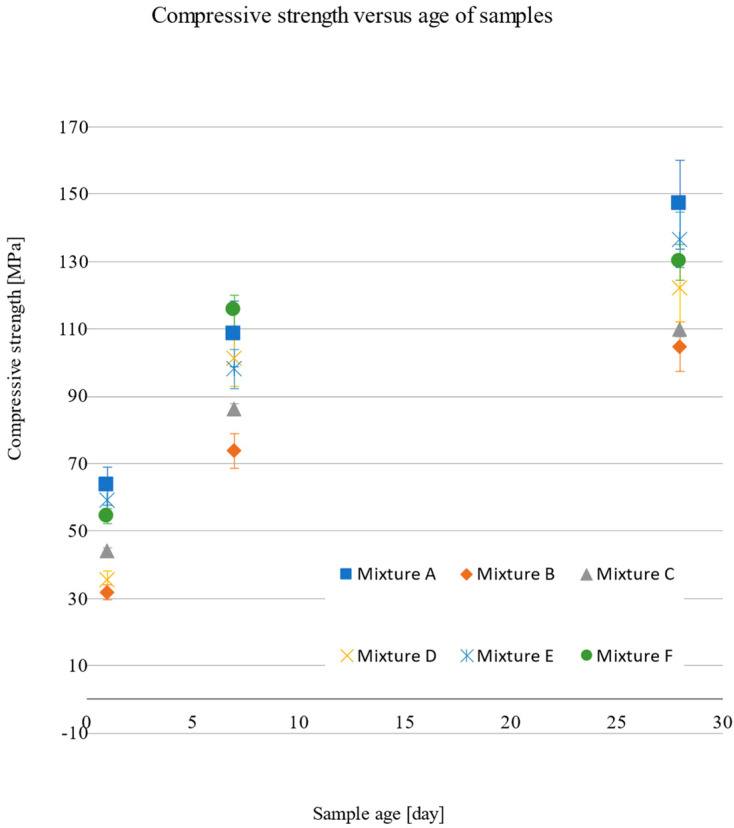
Compressive strength as a function of the age of the sample.

**Figure 8 materials-15-05916-f008:**
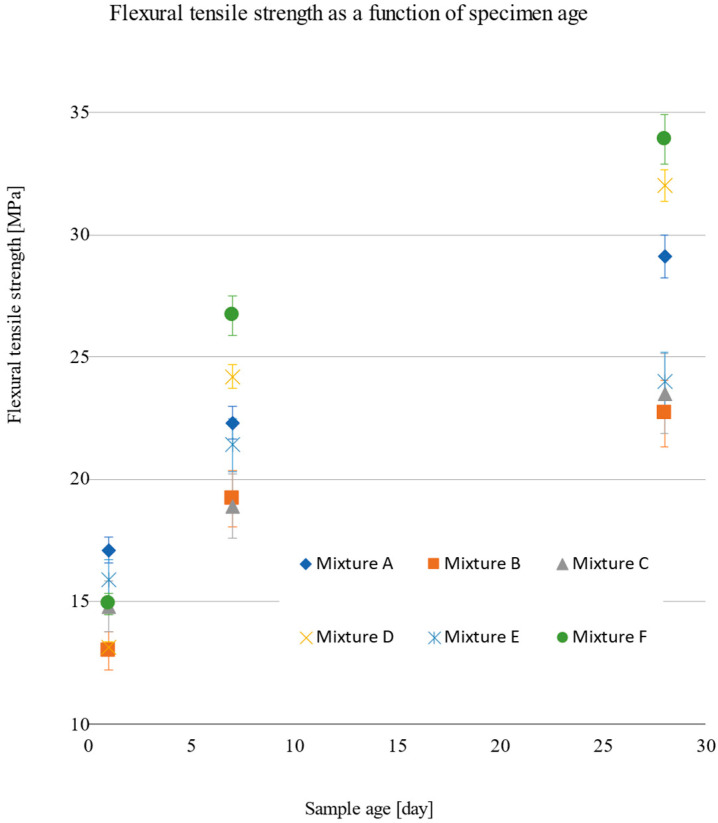
Tensile strength as a function of the age of the sample.

**Figure 9 materials-15-05916-f009:**
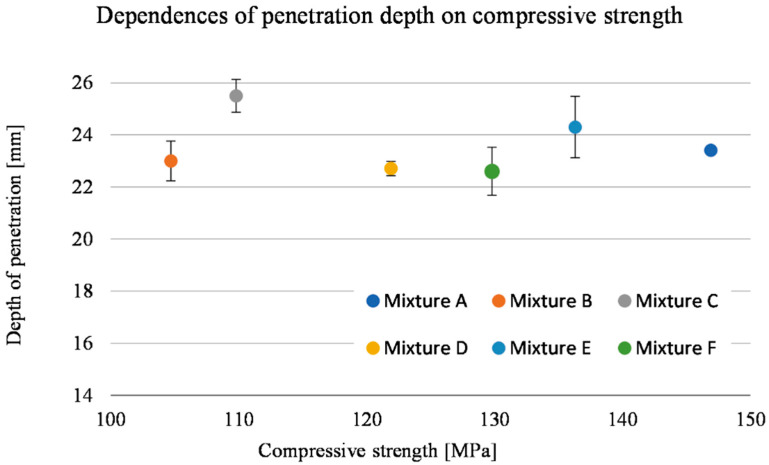
Depth of penetration as a function of compressive strength.

**Figure 10 materials-15-05916-f010:**
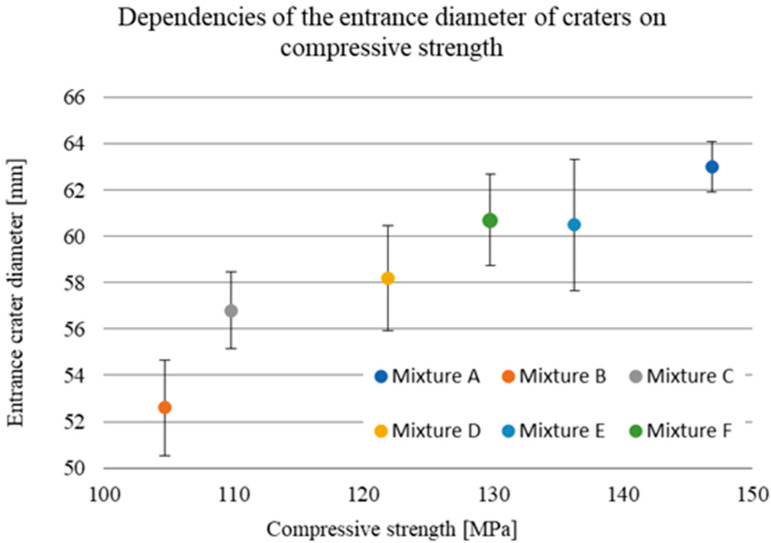
Entrance crater diameter versus compressive strength.

**Figure 11 materials-15-05916-f011:**
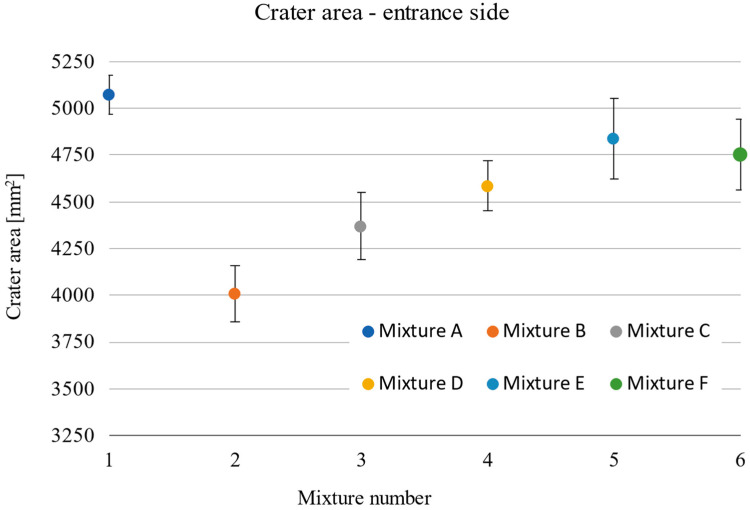
Size of the entrance crater area by individual mixtures.

**Figure 12 materials-15-05916-f012:**
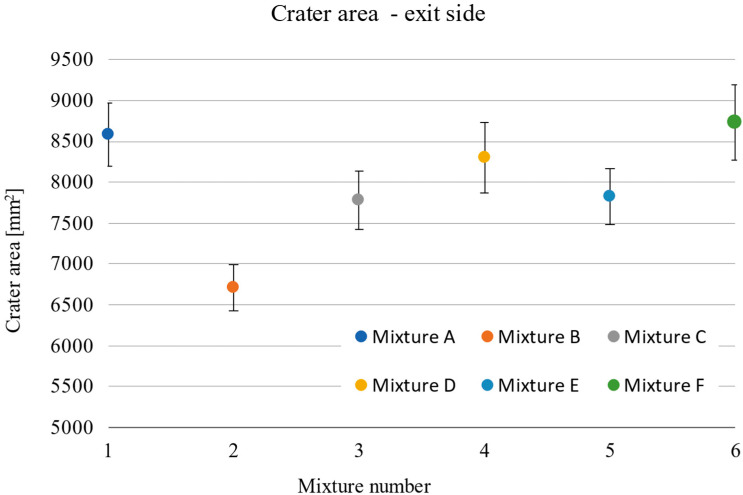
Area of exit crater by individual mixtures.

**Figure 13 materials-15-05916-f013:**
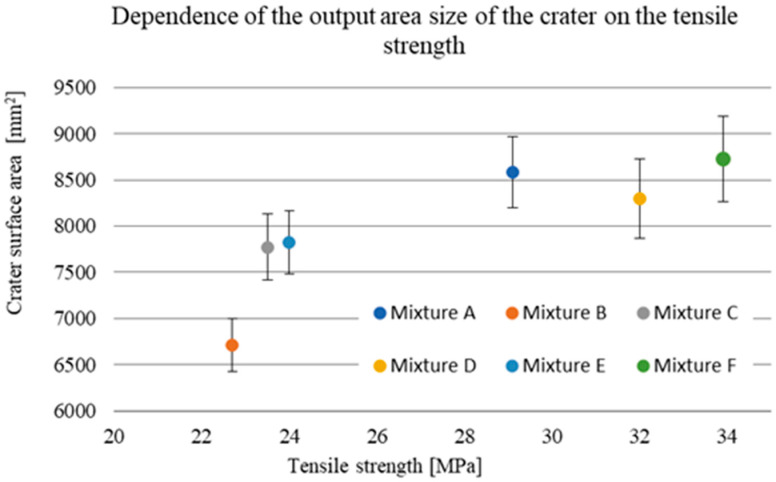
Outlet crater surface size versus compressive strength.

**Table 1 materials-15-05916-t001:** Composition of the reference mixture with a defined aggregate ratio.

Mixture Composition	kg/m^3^	Weight Ratios
Cement CEM I 52.5 R	695	1
Admixtures	270	0.39
Fine sands	1225	1.76
Water	165	0.24
Superplasticizers	40	0.06
Steel fibres	120	0.17

**Table 2 materials-15-05916-t002:** Fractions of aggregates, sands, and microsands used in this study.

Type of Filler	Grain Size (mm)
**Aggregates**	
Silica sand	0.0–2.0
**Microsands**	
Microsand 01/06	0.1–0.6
Microsand 03/08	0.3–0.8
Microsand 06/12	0.6–1.2

**Table 3 materials-15-05916-t003:** Rations of aggregates and microsands used for each mix.

Name Composition	Type of Aggregate and Microsand (g/kg)			
	Microsand 01/06	Microsand 03/08	Microsand 06/12	Aggregates 0/2
Mixture A	250.0	500.0	250.0	-
Mixture B	-	333.3	-	666.7
Mixture C	333.3	-	-	666.7
Mixture D	100.0	293.0	-	607.0
Mixture E	105.3	175.4	122.8	596.5
Mixture F	103.4	241.4	120.7	534.5

**Table 4 materials-15-05916-t004:** Properties of UHPSFRC, mixture A.

Curing Time	f_cf_ (MPa)	f_c_ (MPa)	Volumetric Mass (kg/m^3^)
1 day	17.1	63.4	2367
7 days	22.3	108.4	2362
28 days	29.1	146.9	2372

**Table 5 materials-15-05916-t005:** Properties of UHPSFRC, mixture B.

Curing Time	f_cf_ (MPa)	f_c_ (MPa)	Volumetric Mass (kg/m^3^)
1 day	13.0	31.7	2277
7 days	19.2	73.7	2312
28 days	22.7	104.7	2270

**Table 6 materials-15-05916-t006:** Properties of UHPSFRC, mixture C.

Curing Time	f_cf_ (MPa)	f_c_ (MPa)	Volumetric Mass (kg/m^3^)
1 day	14.8	44.0	2334
7 days	18.9	86.1	2312
28 days	23.5	109.8	2270

**Table 7 materials-15-05916-t007:** Properties of UHPSFRC, mixture D.

Curing Time	f_cf_ (MPa)	f_c_ (MPa)	Volumetric Mass (kg/m^3^)
1 day	13.1	35.3	2405
7 days	24.2	101.0	2371
28 days	32.0	121.9	2358

**Table 8 materials-15-05916-t008:** Properties of UHPSFRC, mixture E.

Curing Time	f_cf_ (MPa)	f_c_ (MPa)	Volumetric Mass (kg/m^3^)
1 day	15.9	59.2	2381
7 days	21.4	98.0	2318
28 days	24.0	136.3	2370

**Table 9 materials-15-05916-t009:** Properties of UHPSFRC, mixture F.

Curing Time	f_cf_ (MPa)	f_c_ (MPa)	Volumetric Mass (kg/m^3^)
1 day	14.9	54.3	2414
7 days	26.7	115.4	2325
28 days	33.9	129.8	2339

**Table 10 materials-15-05916-t010:** Damage sizes of UHPSFRC samples.

Name Composition	DOP (mm)		Diameter of the Entrance Crater (mm)		Crater Area (mm^2^)			
	Average Value	Standard Deviation	Average Value	Standard Deviation	Entrance Side		Exit Side	
					Average Value	Standard Deviation	Average Value	Standard Deviation
Mixture A	23.4	0.1	63.0	1.1	5072	108	8582	381
Mixture B	23.0	0.8	52.6	2.1	4011	151	6712	282
Mixture C	25.5	0.6	56.8	1.7	4371	179	7773	357
Mixture D	22.7	0.3	58.2	2.3	4585	133	8299	428
Mixture E	24.3	1.2	60.5	2.8	4838	218	7828	341
Mixture F	22.6	0.9	60.7	2.0	4754	190	8728	463

DOP = Depth of penetration (mm). Crater area is the actual area of the crater as read from a 3D scan of the crater. It is not just a plan projection of the area.

## Data Availability

The data presented in this study are available on request from the corresponding author.

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
