# Peer review of "Effect of the Microsand Fraction on the Ballistic Resistance of UHP-SFRC"

_materials, 2022, doi:10.3390/ma15175916_

Round 1
Reviewer 1 Report
This is an interesting paper dealing with the ballistic resistance of UHP-SFRC, offering a series of basic data on how the experimental parameters affect the physical and mechanical properties. It's well organized and structured. There are some issues needed to be clarified before it can be accepted for publication in Materials:
1. The manuscript should be very carefully checked to avoid any errors. The language should be checked throughout the text and any grammar mistakes should be corrected. The tenses in the manuscript are confusing and the reviewers suggest that the authors revise carefully.
2. The authors should explain or annotate the abbreviations at their first use site. It is beneficial to readers who are not familiar with this field. E.g. 2.1. Reference Samples UHP-SFRC?
3. The introduction is too short and the author must expand it. There are many studies on UHP-SFRC and the authors should conduct a systematic review. E.g. Science of The Total Environment, 836, 2022, 155424. Powder Technology 398, 2022, 117075.
4. Table 1. What is the reason of choosing those W/C ratios and the percentage of SP? Please explain it in the manuscript.
5. Error bars should be entered on all graphs. This is the only way to ensure that the results obtained are reliable and the trends obtained are valid. It is also advisable to introduce some type of statistical analysis such as ANOVA to provide reliability to the conclusions obtained.
6. Fig. 9. How did the authors judge CH, CSH and unhydrated particles? This is obviously unfounded.
7. The theoretical analysis needs a more in-depth discussion.
Author Response
Comments and Suggestions for Authors
This is an interesting paper dealing with the ballistic resistance of UHP-SFRC, offering a series of basic data on how the experimental parameters affect the physical and mechanical properties. It's well organized and structured. There are some issues needed to be clarified before it can be accepted for publication in Materials:
A: Thank you for your comments and suggestions. We hope that our editing has helped to improve the article.
- The manuscript should be very carefully checked to avoid any errors. The language should be checked throughout the text and any grammar mistakes should be corrected. The tenses in the manuscript are confusing and the reviewers suggest that the authors revise carefully.
A: Thank you for the notice.
- The authors should explain or annotate the abbreviations at their first use site. It is beneficial to readers who are not familiar with this field. E.g. 2.1. Reference Samples UHP-SFRC?
A: Thank you for the notice, explanation added to the text
- The introduction is too short and the author must expand it. There are many studies on UHP-SFRC and the authors should conduct a systematic review. E.g. Science of The Total Environment, 836, 2022, 155424. Powder Technology 398, 2022, 117075.
A: The introduction has been extended.
- Table 1. What is the reason of choosing those W/C ratios and the percentage of SP? Please explain it in the manuscript.
A: The explanation was adding to the chapter 2.1. Reference Samples UHP-SFRC.
- Error bars should be entered on all graphs. This is the only way to ensure that the results obtained are reliable and the trends obtained are valid. It is also advisable to introduce some type of statistical analysis such as ANOVA to provide reliability to the conclusions obtained.
A: The data were processed according to statistical rules. However, there is not enough data for a more extensive statistical analysis. Testing more samples was unfortunately, beyond the time possibilities of the work.
- Fig. 9. How did the authors judge CH, CSH and unhydrated particles? This is obviously unfounded.
A: The aim of this work was the macroanalysis of ballistic damage. Microanalysis was not part of this work. But of course, this is another topic for ongoing research.
- The theoretical analysis needs a more in-depth discussion.
A: Discussion has been edited.

Reviewer 2 Report
1.The presented purpose of the study is vague and more states the process itself. The authors need to specify the ultimate goal of their research and identify the tasks to be solved. Authors are suggested to improve the introduction part and use more literatures.
2.Please explain how did they design mixtures A to F in table 3.
3. Why authors did not use mixture with only 333.3(g/kg) ST 06/12.
4. Please provide more details in figure 3
5. line 117 "lower than their impact velocity " should be "lower than their muzzle velocity"
6. It would be great if authors could present the 3D model of damaged area.
7. Static tests result (table4 to 9 and fig 5 and 6) just discussed in a short paragraph, please explain and discuss more about this result.
8. It would be great if authors could provide failure image of samples during the static test.
9. authors are suggested to discuss more about fracture mechanism and cracks propagation in samples
10. Image of the samples after ballistic test should be added
11. It will be important to determine relation between the compressive strength and carter size through statistical analysis.
12. It would be great to discuss more about the penetration process in the sample, shock wave propagation in sample.
Author Response
Comments and Suggestions for Authors
A: Thank you for your comments and suggestions. We hope that our editing has helped to improve the article.
1.The presented purpose of the study is vague and more states the process itself. The authors need to specify the ultimate goal of their research and identify the tasks to be solved. Authors are suggested to improve the introduction part and use more literatures.
A: The introduction has been extended.
2.Please explain how did they design mixtures A to F in table 3.
A: The explanation was adding to the chapter 2.1. Reference Samples UHP-SFRC.
- Why authors did not use mixture with only 333.3(g/kg) ST 06/12.
A: If we used a mixture with only 333.3 (g/kg) ST 06/12, the content of smaller fractions (0 – 0.57 mm) would decrease and this would negatively affect mechanical and ballistic parameters (see results of mixture B and mixture C).
- Please provide more details in figure 3
A: Picture expanded with side view.
- line 117 "lower than their impact velocity " should be "lower than their muzzle velocity"
A: Yes, thank you for the notice.
- It would be great if authors could present the 3D model of damaged area.
A: 3D scan image added (Fig. 6).
- Static tests result (table4 to 9 and fig 5 and 6) just discussed in a short paragraph, please explain and discuss more about this result.
A: Added a short summary.
- It would be great if authors could provide failure image of samples during the static test.
A: We apologize, but photos and footage were only taken of samples that were ballistically tested. No photo of static loading was taken.
- authors are suggested to discuss more about fracture mechanism and cracks propagation in samples
A: The text has been expanded in the introduction and discussion to include a description of the damage mechanism.
- Image of the samples after ballistic test should be added
A: Added image of sample after ballistic test (Fig.5).
- It will be important to determine relation between the compressive strength and carter size through statistical analysis.
A: The data were processed according to statistical rules. However, there is not enough data for a more extensive statistical analysis.
- It would be great to discuss more about the penetration process in the sample, shock wave propagation in sample.
A: The text has been expanded in the introduction and discussion to include a description of the damage mechanism.

Round 2
Reviewer 1 Report
Although the authors have done their best to revise the manuscript, I do not think this manuscript is suitable for publication in its current form.
1. There are many researches on UHP-SFRC, what makes this research different?
2. The manuscript should be very carefully checked to avoid any errors. The language should be checked throughout the text and any grammar mistakes should be corrected.
3. The authors should explain or annotate the abbreviations at their first use site. It is beneficial to readers who are not familiar with this field. E.g. ST in Table 3. If the manuscript remains unrevised, I have to reject it.
4. Figs. 2, 5 and 6. Please add a scale bar.
5. Aside from the aim stated in the title, the research gap and the goals of the research are not specified which leads to the reader missing the significance of the research.
6. Methods section determines the results. Kindly focus on three basic elements of the methods section. a. How the study was designed? b. How the study was carried out? c. How the data were analyzed?
7. Please make sure your conclusions' section underscore the scientific value added of your paper, and/or the applicability of your findings/results, as indicated previously. Please revise your conclusion part into more details. Basically, you should enhance your contributions, limitations, underscore the scientific value added of your paper, and/or the applicability of your findings/results and future study in this session.
8. Error bars should be entered on all graphs. This is the only way to ensure that the results obtained are reliable and the trends obtained are valid. It is also advisable to introduce some type of statistical analysis such as ANOVA to provide reliability to the conclusions obtained.
9. The theoretical analysis needs a more in-depth discussion. The underlying mechanism has not been explained.
10. The research contributions of the paper should be articulated more clearly. The abstract is not representative of the content and contributions of the paper. The abstract does not seem to properly convey the rigor of research.
